# The burden of disabilities in Sidama National Regional State, Ethiopia: A cross-sectional, descriptive study

Zelalem Tenaw[1]*, Taye Gari[2], Achamyelesh Gebretsadik[2]

1 Department of Midwifery, College of Medicine and Health Sciences, Hawassa University, Hawassa, Ethiopia, 2 School of Public Health, College of Medicine and Health Sciences, Hawassa University, Hawassa, Ethiopia

* abigiatenaw@gmail.com

## Abstract

**Data Availability Statement:** All relevant data are within the paper and its Supporting Information files.

**Funding:** The author(s) received no specific funding for this work.

### Background

Assessing the burden and describing the status of people with disabilities is very essential. The previous studies conducted about the prevalence, causes, and types of disability in Ethiopia were inconsistent and disagreeable.

### Objectives

To determine the prevalence, causes, and types of disabilities in Sidama National Regional State, Ethiopia.

### Methods

A house-to-house census was carried out on a total of 39,842 households in 30 randomly selected kebeles of the Dale and Wonsho districts and Yirgalem city administration, Sidama National Regional State. The data were collected using structured and pretested questionnaires via the Kobo Collect application from May 01 to 30, 2022. The analysis was performed by STATA version 16 software. After cleaning and organizing, descriptive statistics were employed to characterize the study findings.

### Results

In this study, people with disabilities aged one to 80 years old were included. The mean Standard Deviation (SD) age of people with disabilities in years was 31.95 (15.33). Of 228,814 people, 1,694 were people with disabilities in Dale and Wonsho districts and Yirgalem city administration, with a prevalence of 0.74% (95% CI: 0.72, 0.76). Of the causes of disability, 61% of the disabilities were due to illness, injury, and accidents. Extremity paralysis (35.4%), vision disability (20.13%), hearing disability (19.7%), walking disability (14.7%), and cognitive disabilities (7.7%) were the identified types of disabilities.

**Competing interests:** The authors have declared that no competing interests exist.

## Conclusion

This study revealed that the burden of disability is considerable in Dale and Wonsho districts and Yirgalem city administration. The vast majority of disability causes could have been avoidable. As a result, developing and implementing various strategies to raise community awareness about the causes and preventive measures is critical.

## Introduction

Although the definition of disability varies from country to country [1,2], we considered people with disability as defined by the World Health Organization (WHO) as; any impairment of a person's body function or structure or mental functioning, activity limitation, and participation restriction (environmental factors) [3]. Globally, 15% of the world population lives with disabilities, of which 75%-80% are from developing countries [4,5]. Although the majority of people with disabilities are from developing countries, their prevalence varies among and within countries. The prevalence of people with disabilities were varied across developing countries, which is supported by the findings in Bangladesh (10.5%) [6], in Kenya (2.2%) [7], in South Africa (9.7%) [8], and in Ethiopia, 1.8% [9] to 4.9% [10]. According to the Ethiopian population and housing census conducted in 2007, the prevalence of disabilities was 1.2% [11] at the national level and 0.85% in Sidama National Regional State. However, the WHO estimates that 17.6% of Ethiopian people are living with disabilities [4].

The causes of disability are multifactorial. Of the causes of disability, 37% are due to illness [12] and 17.8% to 35.9% are due to injury [9,10,12], and infections like meningitis, measles, maternal rubella, and poliomyelitis [10,12] were some of the causes of disability. On the other hand, different studies revealed different types of disabilities in Ethiopia, including 34% to 51% vision disability [9,10,12], 11.7% to 22.3% walking disability [9,10,12], 6.8% to 22.3% hearing disability [9,10,12] and 15% cognitive disability [9,12].

The prevalence of disabilities in Ethiopia is very inconsistent, ranging from 1.8% [9] to 4.9% [10]. The prevalence of 1.2% from the Ethiopian population and housing census prevalence was very late (conducted 15 years back) [11]. There is also a significant disparity between World Health Organization estimates of disabled people and the actual number of disabled people. Likewise, there is a disagreement between the findings regarding the causes and types of disability in Ethiopia, evidenced by a 17.8% to 35.9% [9,10,12] difference in the magnitude of disability due to injury and a 34% to 51% [9,10,12] difference in the types of vision disability.

Therefore, determining the prevalence of people with disabilities and identifying the causes and types of disabilities is very important for policymakers, program planners, and researchers to fill the knowledge gap and design different strategies to reach people with disabilities based on the magnitude, causes and types of disabilities in Sidama National Regional State, Ethiopia.

## Methods and materials

### Study design and setting

A house-to-house census was conducted from May 01 to 30, 2022 to determine the prevalence, causes, and types of disabilities in Sidama National Regional State, Ethiopia. The research was carried out in the Dale and Wonsho districts, as well as the Yirgalem city administration. According to the Sidama National Regional State Report (2021/2022), the total population of

Dale district was 254,653; that of Wonsho district was 129,730, and Yirgalem city administration was 85,072 [13,14] with an estimated 53,768 households (HHs) in Dale district, 21,857 households in Wonsho district, and 42,902 households in Yirgalem city administration. The two districts are the Health and Demographic Surveillance sites of Hawassa University. Both districts are known for their coffee production and highly dense populations and are considered representative of the region's population of their socio-economic and cultural issues. Dale district has 36 rural and two urban kebeles (lower political administrative units in Ethiopia), while Wonsho district has 17 rural and two urban kebeles, and Yirgalem city administration has three rural and six urban kebeles.

### Estimated sample size (households)

The estimated households for the census were 39,420.

### Sampling technique and procedure

The study was conducted in randomly selected urban and rural kebeles in the Dale and Wonsho districts and the Yirgalem city administration. There are 56 rural and 10 urban kebeles in Dale, Wonsho districts, and Yirgalem city administration. Of these kebeles, 30 kebeles (20 rural and 10 urban) were selected randomly. A multi-stage stratified random sampling technique was used. The stratification was based on residency (rural and urban).

### Data collection procedure

Six experienced BSc Nurses who are fluent in the Sidamu Afoo language collected data and were supervised by one experienced supervisor (MSc holder). The data collectors visited the selected kebeles and created community awareness about the purpose of the census by collaborating with the kebele's health extension worker and the manager of the kebele and assembling the community in the kebele's venue before starting the census data collection. The assembling was done to minimize the informational bias of the head or leader of the household. Then, a house-to-house census among the eligible and selected kebeles was conducted to identify people with disabilities. The heads of the households were asked about the presence of a person who has a problem with seeing, hearing, speaking, and/or standing, walking, or sitting, body part movement, the functioning of hands and legs, or mental retardation or mental problems among the family members that confirmed by medical staffs. When there were people with disabilities in the household, other important variables, including sociodemographic characteristics, types of disability, and causes of disability, were asked. The data collection tool (questionnaire) was adopted from the 2007 Ethiopian Central Statistical Agency (CSA) population and housing census [11]. We obtained permission to download and use the survey data on September 27, 2021.The data collection tool was prepared in English and then translated into the Sidamu Afoo language and translated back to English to check and retain its originality and consistency. The questionnaire was pretested in two (one urban and one rural) kebeles in the Hawassa city administration.

### Data analysis

The data were collected using the Kobo collect application and imported into Stata version 16 for analysis using the "SSC install kobo2stata" command. The details of the data analysis were described elsewhere [15]. To describe the characteristics of people with disabilities, we employed descriptive statistics like percentage and mean with standard deviation. The prevalence was calculated by considering the 2021–2022 latest Kebeles population report as a

denominator because, because of resource limitations, we could not enumerate the total number of people in the households.

## Ethical considerations

The ethical clearance was obtained from the Institutional Review Board at the College of Medicine and Health Sciences of Hawassa University with an approval number of ref. no. IRB/143/14. After approval, a support letter was written to the Sidama National Regional Public Health Institute. Then, after obtaining the support letter from Sidama National Regional Public Health Institute, the permission and cooperation letter were given to the woreda health offices. Finally, the woreda health offices wrote a permission letter to selected kebeles, asking them to cooperate and give consent to conduct the study. Verbal consent was gained from the head of household. There was a yes or no question in the tool and the data collectors tick whether the study participants are volunteer or not with the presence of kebele (smallest administrative area) leader to collect the data. There is no risk in participating in this survey. People with disabilities having different health problems were linked to nearby health facilities for possible support and follow-up. The participants were recruited from May 01 to 30, 2022.

## Results

### Socio-demographic characteristics of people with disabilities

A total of 39,842 households with a total population of 228,814 were included in this study from 30 rural and urban kebeles. Of these populations, a total of 1694 people with disabilities were identified and included in this study. The mean (SD) age of people with disabilities was 31.95 (15.33) years. Among people with disabilities, 999 (58.97%) were male and 1114 (65.57%) resided in rural settings. Of the study participants, 1044 (61.63%) were not able to read and write (Table 1).

### Prevalence of people with disabilities

A total of 39,842 households with a total population of 228,814 were included in this study. Of these populations, 1,694 were people with disabilities, which corresponds to an overall prevalence rate of 0.74% (95% CI: 0.72, 0.76). Of the total households, 4.3% (95% CI: 4%, 4.5%) reported at least one member with a disability.

### Reported causes of disabilities

As listed in Fig 1, different causes were mentioned by people with disabilities or other concerned bodies as causes of disability in the Dale and Wonsho districts and Yirgalem city administration. The most common causes are inborn 614 (36%) and illness 598 (35%) (Fig 1).

### Types of disabilities

Of 1694 people with disabilities, more than one-third (35.4%) had severe paralysis/handicap disability, followed by vision disability 341(20.1%). Among the people with disabilities, 41 (2.4%) had more than one type of disability (Fig 2).

### Types of disability by age

Regardless of gender, all types of disabilities were more prevalent among people in the productive age group (16 to55 years old) (Table 2).

**Table 1. Socio-demographic characteristics of people with disabilities in Sidama Regional Stata, Ethiopia, 2022.**

| Variable | | Number | Percent |
|---|---|---|---|
| Age in years | Mean (SD) | 31.95 (15.33) | |
| Sex | Female | 695 | 41.03 |
| | Male | 999 | 58.97 |
| Marital status | Never married | 1017 | 60.04 |
| | Married | 622 | 36.72 |
| | Widowed | 46 | 2.72 |
| | Divorced | 9 | 0.53 |
| Residence | Rural | 1114 | 65.57 |
| | Urban | 580 | 34.24 |
| Employment | Employed | 35 | 2.63 |
| | Not employed | 1659 | 97.93 |
| Educational status | Unable to read and write | 1044 | 61.63 |
| | Primary education | 398 | 23.5 |
| | Secondary education | 233 | 13.75 |
| | College/University | 19 | 1.43 |
| Disabilities association member | Yes | 409 | 24.2 |
| | No | 1281 | 75.8 |

## Causes of disability by age

For both genders, all causes of disabilities were prevalent among people in the productive age group (16 to 55 years old) (Table 3).

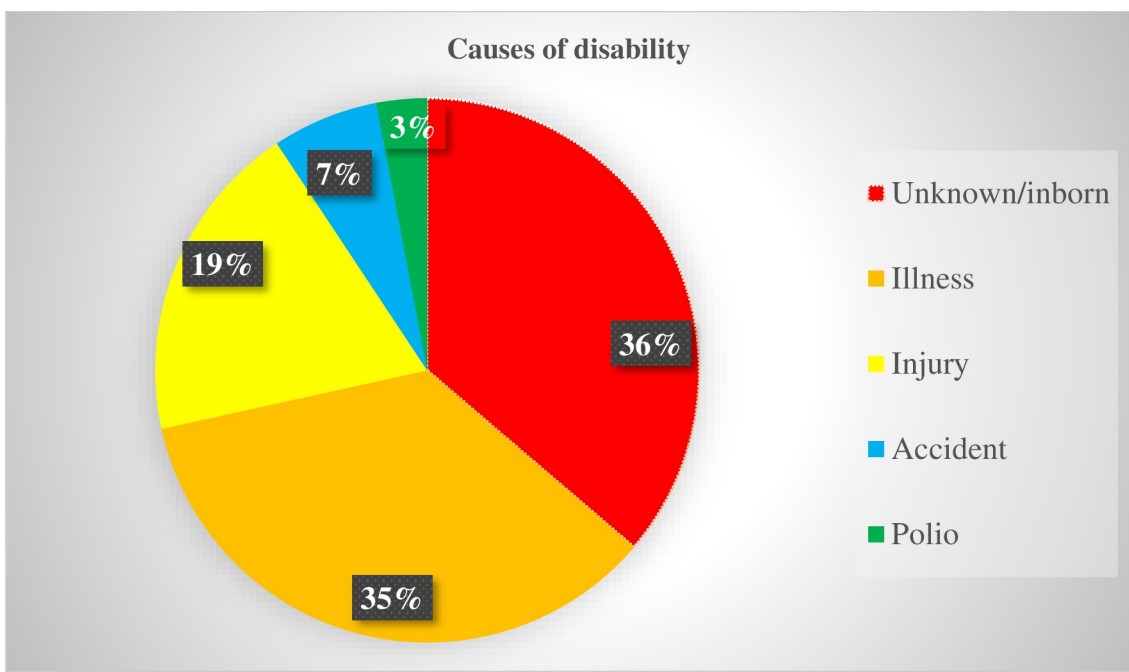

**Fig 1. Causes of disability in Dale and Wonsho districts and Yirgalem city administraton, Sidama National Regional State, Ethiopia, 2022.**

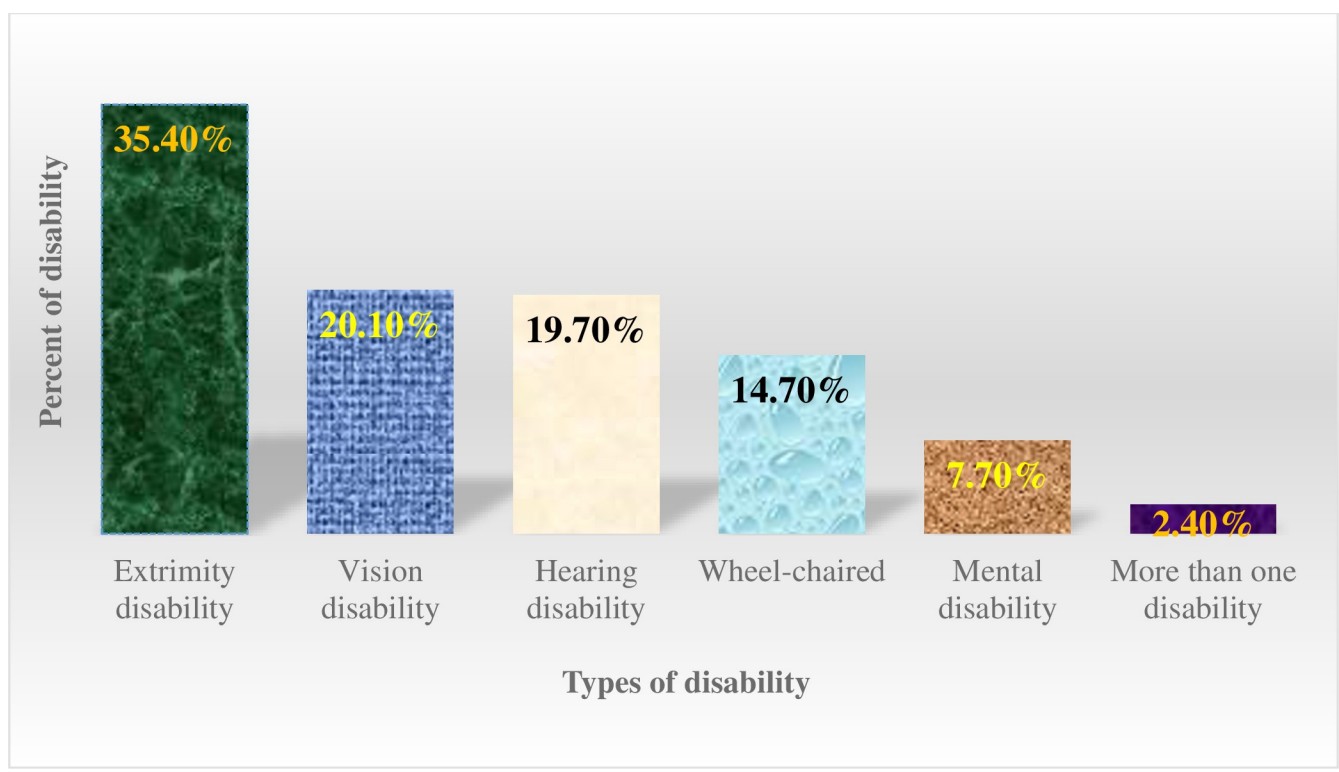

**Fig 2. Types of disability in Dale and Wonsho districts and Yirgalem city administration, Sidama National Regional State, Ethiopia, 2022.**

## Discussion

The prevalence of people with disabilities in Dale and Wonsho districts and Yirgalem city administration, Sidama National Regional State, Ethiopia was 0.74%. Inborn, illness, injury, accidents and polio are the identified causes of disability. Extremity paralysis, vision disability, hearing disability, walking disability, and cognitive disabilities were the types of disabilities in the region.

This study's prevalence is lower than studies conducted in Bangladesh (10.5%) [6], South Africa (9.7%) [8] and Kenya (2.2%) [7]. The possible reasons might be that all three studies

**Table 2. Types of disability by age in Dale and Wonsho districts and Yirgalem city administration, Sidama National Regional State, Ethiopia, 2022.**

| Types of disability (n = 1694) | Age 1 to 5 years | Age 6 to 15 years | Age 16 to 25 years | Age 26 to 35 years | Age 36 to 45 years | Age 46 to 55 years | Age 56 to 65 years | Age 66 to 80 years |
|---|---|---|---|---|---|---|---|---|
| Extremity (n = 599) | 3(0.16%) | 35(2.07%) | 133(7.85%) | 179(10.57%) | 122(7.20%) | 79(4.66%) | 33(1.95%) | 15(0.89%) |
| Vision (n = 341) | 4(0.24%) | 17(1.00%) | 76(4.49%) | 56(3.31%) | 61(3.60%) | 64(3.79%) | 35(2.07%) | 28(1.65%) |
| Hearing (n = 334) | 3(0.16%) | 44(2.60%) | 106(6.26%) | 68(4.01%) | 47(2.77%) | 39(2.30%) | 11(0.65%) | 16(0.94%) |
| Walking (n = 249) | 6(0.35%) | 17(1.00%) | 61(3.60%) | 86(5.08%) | 40(2.36%) | 29(1.79%) | 7(0.41%) | 3(0.16%) |
| Mental (n = 130) | 0(0%) | 13(0.89%) | 45(2.66%) | 42(2.48%) | 19(1.12%) | 5(0.30%) | 2(0.12%) | 4(0.24%) |
| Multiple (n = 41) | 3(0.18%) | 1(0.06%) | 10(0.59%) | 10(0.59%) | 12(0.71%) | 3(0.18%) | 1(0.06%) | 1(0.06%) |
| Total | 19(1.1%) | 127(7.5%) | 431(25.4%) | 441(26.0%) | 301(17.8%) | 219(12.9%) | 89(5.3%) | 67(4.0%) |

**Table 3. Causes of disability by age in Dale and Wonsho districts and Yirgalem city administration, Sidama National Regional State, Ethiopia, 2022.**

| Causes of disability | Age 1 to 5 years | Age 6 to 15 years | Age 16 to 25 years | Age 26 to 35 years | Age 36 to 45 years | Age 46 to 55 years | Age 56 to 65 years | Age 66 to 80 years |
|---|---|---|---|---|---|---|---|---|
| Inborn (n = 614) | 8(0.47%) | 71(4.19%) | 177(10.45%) | 148(8.74%) | 100(5.90%) | 76(4.49%) | 19(1.12%) | 15(0.89%) |
| Illness (n = 598) | 2(0.12%) | 27(1.59%) | 131(7.73%) | 140(8.26%) | 111(6.55%) | 94(5.55%) | 52(3.07%) | 41(2.42%) |
| Injury (n = 324) | 6(0.35%) | 7(0.41%) | 89(5.25%) | 113(6.67%) | 60(3.54%) | 33(1.95%) | 10(0.59%) | 6(0.35%) |
| Accident (n = 107) | 0(0%) | 13(0.77%) | 20(1.18%) | 23(1.36%) | 24(1.42%) | 16(0.94%) | 7(0.41%) | 4(0.24%) |
| Polio (n = 51) | 3(0.18%) | 9(0.53%) | 14(0.83%) | 17(1.00%) | 6(0.35%) | 0(0%) | 1(0.06%) | 1(0.06%) |
| Total | 19(1.12%) | 127(7.2%) | 431(34%) | 441(27.3%) | 301(16%) | 219(9.2%) | 89(3%) | 67(1.4%) |

were excluded under five cases, especially the Bangladesh study, which was considered to have participants over 18 years of age which means disability is prevalent [10] and also the study period (2010 vs 2022) and sample size might be the other possible reasons. As the number of denominators increases, the prevalence will be decreased. The prevalence observed in this study is lower than that reported in studies done in Dabat districts, Ethiopia (4.9% [10], 3.8% [16], and the Dabat health demographic monitoring site (2.14% [12] and 1.8% [9]), which were all conducted in Ethiopia. The discrepancy may be due to the study year 2001, which was the time when the maximum prevalence of 4.9% was reported. The sample size may also be the cause of the discrepancy, as the second maximum prevalence of 3.8% was found in a population of 24,453 people, which is much smaller than the population of this study (228,814). The other possible justification might be the fact that most of the wars in Ethiopia happened in the northern part of the country rather than the southern part of Ethiopia [17]. This might increase the prevalence of disability. In general, when we compare the prevalence of disability with study periods, the prevalence of disability decreased over time, which might be because of the improvements in the medical care system like outreach and static cataract surgery, vaccination, and socio-economic factors [18,19]. The other reason can be best explained by the improvement of the medical health care system in terms of access and service diversification and urgent treatment of disability-causing problems [20].

In this study, the most frequently identified causes of disability in Dale and Wonsho districts and Yirgalem city administrations, Sidama National Regional Stata, Ethiopia are; Inborn/unknown 36%, illness 35%, injury 19%, accidents 7% and polio 3%. This evidence is in line with the study conducted in the Dabat district in Northern Ethiopia, with illness at 37% and injury at 17.8% to 35.9% [9,12]. In this study, 61% of the disabilities are due to illness, injury, and accidents, which are preventable. As evidenced by different findings [12,21] the majority of the disabilities in Africa are due to the above three preventable causes. In low and middle-income countries, including Ethiopia, the majority of the disabilities are caused by different infections like meningitis, measles, maternal rubella and poliomyelitis [10,22] which are avoidable and preventable by early detection and treatment [12]. But due to poor health-seeking behaviour and access, many people are the victims of disabilities in low and middle-income countries, including Ethiopia. By encouraging preconception care and antenatal follow-up, we can also minimize or prevent inborn causes of disability, which is one of the commonest 36% causes of disability, which may be important to prevent congenital anomalies that could cause disability [23,24].

Regarding types of disability, extremity paralysis (35.4%), vision disability (20.13%), hearing disability (19.7%), walking disability (14.7%), and cognitive disabilities (7.7%) were the identified types of disabilities, (2.4%) of people with disabilities had more than one type of disability. This finding is different from the studies conducted in the Dabat district, Ethiopia and North Ethiopia [9,10,12]. In the Dabat study, the leading causes were 51% vision disability, 22.3%

walking disability, and 22.3% hearing disability [12]. The other study from the Dabat health demographic surveillance site identified vision disability (39%), hearing disability (18%), walking disability (17%), and cognitive disability (15%) as the leading types of disability [9]. Another study from Dabat district, North Ethiopia, revealed walking disability, vision disability, and extremity paralysis/handicap were the common types of disability [10]. The Nigerian study, like the Ethiopian studies, discovered that 37% had vision disabilities, 30% had mobility disabilities, 15% had hearing disabilities, and 9% had mental disabilities [22], which could be due to differences in healthcare-seeking practices for specific disabilities caused by relating with different cultural thoughts. In the Dabat district study [12], 8% of people with disabilities had more than one disability. This prevalence is higher than ours and may be due to the sample size difference (denominator) of 71,673 in Dabat and 228,814 in this study.

People with disabilities have the right to access and attend education, and various strategies such as inclusive education have been designed and implemented in low and middle-income countries such as Ethiopia [25,26]. In Dale and Wonsho districts and Yirgalem city administration, Sidama National Regional State, Ethiopia, 1044 (61.6%) cannot write and read. Only 650 (38.4%) attended formal education, with a mean (SD) grade completed. In the region, 398 (23.5%) attended primary and 233 (13.75%) secondary schools. The most important issue here is that only 19 (1.43%) of people with disabilities had completed college or university. According to the evidence, access to and continuation of higher-grade education for formal education attendants is limited in Dale and Wonsho districts and Yirgalem city administrations. This evidence is supported by the study conducted in the Dabat district, North Ethiopia [12]. The reasons might be a negative attitude toward people with disabilities, a lack of educational supportive materials (braille books and papers), and a lack of skills and training for teachers [26]. Of the formal education attendants, 25.1% were male, 13.3% were female, and 20.9% resided in rural settings. Exclusion from education affects the employment status of people with disabilities [22]. In this study, only 35 (2.63%) people with disabilities were employed. It indicates they cannot generate income and are economically dependent since employment is one of the sources of income [22].

The other important finding of this study is assessing the status of disabilities association and membership of the association. Of the 1694 people with disabilities, 409 (24.2%) were disabilities association members. Thirteen percent of the members were urban residents, and of these, 6.5% were males, and 11.2% were from rural settings, of which 7.14% were males. This finding indicates that a majority (75.8%) of people with disabilities were excluded from different services that came through disabilities associations, and the associations in our setting considered all the people with disabilities included in the associations and considered only those who are members of disabilities associations. Disabilities associations play a pivotal role in evaluating and monitoring different services given and needed for people with disabilities [27].

This study is the first in Sidama National Regional State, Ethiopia to characterize and investigate the burden of disability in the region, which could be used to fill the knowledge gap and may be important to program planners, policymakers, and researchers. To minimize information bias from the head of the household, awareness creation for the community about the importance of the census, involving kebele's health extension workers and managers during the data collection, was done. The other strength of this study was considering the large sample size (household and population). However, due to the large sample size and resource limitations, we as researchers could not consider disability screening methods in the data collection (especially mental retardations and problems). The other limitation of this study was taking of denominator from the kebeles registration to determine the prevalence of people with disabilities, this might affect the prevalence of people with disabilities in the study districts and city administration.

## Conclusions

This study revealed that the burden of disability is considerable in Dale and Wonsho districts and Yirgalem city administration, Sidama National Regional State, though the prevalence is lower than the previous studies conducted in Ethiopia. The majorities of the causes of disabilities were avoidable. Therefore, community awareness about the causes and preventive strategies is important.

## Supporting information

**S1 File.**
(PDF)

**S2 File. Questionnaires (English version).**
(DOCX)

**S3 File. Questionnaires (Afo-sidamu).**
(DOCX)

**S1 Raw data.**
(DTA)

## Acknowledgments

We would like to thank all the household leaders who participated in this study for their genuine responses about the presence of people with disabilities in the household. Lastly, we are also thankful to the Sidama National Regional State Health Department and selected woredas and kebeles for their help and permission to undertake the research.

## Author Contributions

**Conceptualization:** Zelalem Tenaw.

**Data curation:** Zelalem Tenaw, Taye Gari, Achamyelesh Gebretsadik.

**Formal analysis:** Zelalem Tenaw, Taye Gari, Achamyelesh Gebretsadik.

**Investigation:** Zelalem Tenaw, Taye Gari, Achamyelesh Gebretsadik.

**Methodology:** Zelalem Tenaw, Taye Gari, Achamyelesh Gebretsadik.

**Project administration:** Zelalem Tenaw.

**Software:** Zelalem Tenaw, Taye Gari, Achamyelesh Gebretsadik.

**Supervision:** Zelalem Tenaw, Taye Gari, Achamyelesh Gebretsadik.

**Validation:** Zelalem Tenaw, Taye Gari, Achamyelesh Gebretsadik.

**Writing – original draft:** Zelalem Tenaw.

**Writing – review & editing:** Zelalem Tenaw.

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
