## [Decision Letter · Decision Letter 0]

23 Jun 2023

PONE-D-22-33207The Burden of Disabilities in Sidama National Regional State, Ethiopia: A Cross-Sectional, Descriptive StudyPLOS ONE

Dear Dr. Tenaw,

Thank you for submitting your manuscript to PLOS ONE. After careful consideration, we feel that it has merit but does not fully meet PLOS ONE’s publication criteria as it currently stands. Therefore, we invite you to submit a revised version of the manuscript that addresses the points raised during the review process.

We look forward to receiving your revised manuscript.

Kind regards,

Chung-Ying Lin

Academic Editor

PLOS ONE

Journal Requirements:

2. In the ethics statement in the Methods, you have specified that verbal consent was obtained. Please provide additional details regarding how this consent was documented and witnessed, and state whether this was approved by the IRB.

Reviewers' comments:

Reviewer's Responses to Questions

**Comments to the Author**

1. Is the manuscript technically sound, and do the data support the conclusions?

Reviewer #1: Yes

2. Has the statistical analysis been performed appropriately and rigorously? 

Reviewer #1: Yes

3. Have the authors made all data underlying the findings in their manuscript fully available?

Reviewer #1: Yes

4. Is the manuscript presented in an intelligible fashion and written in standard English?

Reviewer #1: No

5. Review Comments to the Author

Reviewer #1: The aticle titled “The Burden of Disabilities in Sidama National Regional State, Ethiopia: A Cross-Sectional, Descriptive Study” has been reviewed. Overall, this is an interesting study while the data will be benefeicial for the policy maker as it use a large sample size. However, there are some issues need to be clarified as follows:

1.In the background, the authors stated the prevalence of disabilities were varied across the countries. It may sound normally as the various character found in each country may resulted in different prevalence. Then, why did the author stated this reason to identify the significance of study? As this is an original study (not the meta-analysis), the results may showed a prevalence that may be different from others again. The gap analysis should be more clear and described in this section.

2.How was the calculation of sample size? What is the reference used to calculate it? Is it based on the previous study or using some statistical analysis such as G power analysis.

3.In the data collection procedure, the authors mentioned “…gathering the community in the kebele’s venue”. What does it mean here? What was the purpose of this procedure?

4.The table 1, the authors provide the data of “widowed” and “divorced”. It may overestimates as widowed may be one type of divorced. Please use more clear and sharp terminology. In addition, the same case with the term “non-government” and “private”. These may cause bias. The

5.Table 2, why did the authors categorize the prevalence of disability into several age category? Is there any reference showing the significance different among those categories? In addition, if so, please add it on the research aim.

6.In the discussion section, the authors described that the exclusion of under-five years participants may cause the difference finding. Please add the reference why the under-five years participants may contribute to the difference. Is there any study supported?

7.In the discussion section, paragraph 6, “The other important finding of this study is assessing the status of disability associations and membership of disabilities”. What does the meaning of disability association? Why is it important?

8.Please do a proof read to ensure each sentence provide clear and sharp meaning.

6. PLOS authors have the option to publish the peer review history of their article (what does this mean?). If published, this will include your full peer review and any attached files.

Reviewer #1: **Yes: **Iqbal Pramukti, Ph.D.

---

## [Author Response · Author response to Decision Letter 0]

30 Jun 2023

Date: 30/06/2023

Response letter

Dear editors

We would like to thank the reviewers and the editor for their invaluable comments, suggestions and their time.

We addressed all the issues raised by the reviewers and editor using their questions and responses, using blue for each comment and question one by one.

We believe the manuscript has been modified based on the given comments and is suitable for publication in PLOS ONE journal.

Zelalem Tenaw (Assistant Professor of Maternity and Rh) 

Editor’s comments (additional requirements) and responses

Response: Dear editor, thank you very much. The manuscript meets PLOS ONE’s style. 

2. In the ethics statement in the Methods, you have specified that verbal consent was obtained. Please provide additional details regarding how this consent was documented and witnessed, and state whether this was approved by the IRB.

Response: The study was approved by the Hawassa University IRB. Then the head of the household asked his or her voluntariness; if volunteer, we tick “yes” and continue; if not volunteer, we tick “no” and leave the household. This was witnessed in the presence of the kebele leader (lines 149-151).

Response: Thank you for the comment. We included a separate caption for both figures.

4. Please include captions for your Supporting Information files at the end of your manuscript, and update any in-text citations to match accordingly.

Response: Thank you again for the comment. We included a caption for both the supporting information.

Response: Thank you for the comments. We reviewed our references and ensured they are complete and correct. And also, we have not cited retracted papers. 

Reviewer 1 comments and responses

1. In the background, the authors stated the prevalence of disabilities were varied across the countries. It may sound normally as the various character found in each country may resulted in different prevalence. Then, why did the author stated this reason to identify the significance of study? As this is an original study (not the meta-analysis), the results may show a prevalence that may be different from others again. The gap analysis should be clearer and more described in this section.

Response: Dear reviewer, thank you for the very constructive comment. We modified the ambiguous statement (see lines 60-62). Otherwise, the gap is clearly showed (see lines 74-81). 

2. How was the calculation of sample size? What is the reference used to calculate it? Is it based on the previous study or using some statistical analysis such as G power analysis.

Response: Thank you again for your comment. We selected 30 kebeles from 66 kebeles based on the estimated number of people with disabilities that we used for the other components of this projects, like (1). After randomly selecting the kebeles, we totally censused all the households present in the selected 30 kebeles (20 rural and 10 urban). 

3. In the data collection procedure, the authors mentioned “…gathering the community in the kebele’s venue”. What does it mean here? What was the purpose of this procedure?

Response: Dear reviewer, thank you very much for your observation and comment. We modified the terminology gathering to assembling (see line 114). The purpose of the assembling was done to minimize the informational bias of the head or leader of the household (see lines 115-116).

4. The table 1, the authors provide the data of “widowed” and “divorced”. It may overestimate as widowed may be one type of divorced. Please use more clear and sharp terminology. In addition, the same case with the term “non-government” and “private”. These may cause bias.

Response: Thank you again. In our context, widowed has totally different meaning from divorced. Therefore, we believe that it is better to maintain the category as it is. Regarding types of employers, we removed it to minimize confusion. 

5. Table 2, why did the authors categorize the prevalence of disability into several age category? Is there any reference showing the significance different among those categories? In addition, if so, please add it on the research aim.

Response: Thank you for your comment and concern. We believed that, since the study is descriptive, showing the burden of disability by age category had many importance for different stockholders to design appropriate strategies based on the age categories and it provides additional information

6. In the discussion section, the authors described that the exclusion of under-five years participants may cause the difference finding. Please add the reference because the under-five years participants may contribute to the difference. Is there any study supported?

Response: Thank you for your observation and comment. The evidence is supported with reference number 10. The justification is that, when the under-five included in the denominator, the prevalence decreased and prevalence roses with age (2). 

7. In the discussion section, paragraph 6, “The other important finding of this study is assessing the status of disability associations and membership of disabilities”. What does the meaning of disability association? Why is it important?

Response: Dear reviewer, Thank you again for your comment. Disabilities associations are one of the important associations to reach and support people with disabilities. That is why assessing disabilities association membership is important. 

8. Please do a proof read to ensure each sentence provide clear and sharp meaning.

 Response: Thank you very much your comment. We did a proof reading to improve the language problems of the manuscript. 

1. Tenaw Z, Gari T, Gebretsadik A. Contraceptive use among reproductive-age females with disabilities in central Sidama National Regional State, Ethiopia: a multilevel analysis. PeerJ. 2023;11:e15354.

2. Tamrat YK, Shitaye Alemu, John Moore, Girmaye,. The prevalence and characteristics of physical and sensory disabilities in Northern Ethiopia. journal of disability and rehabilitation. 2001;23(17):799-804.

---

## [Editor Report · Decision Letter 1]

5 Jul 2023

The Burden of Disabilities in Sidama National Regional State, Ethiopia: A Cross-Sectional, Descriptive Study

PONE-D-22-33207R1

Dear Dr. Tenaw,

We’re pleased to inform you that your manuscript has been judged scientifically suitable for publication and will be formally accepted for publication once it meets all outstanding technical requirements.

Kind regards,

Chung-Ying Lin

Academic Editor

PLOS ONE

Additional Editor Comments (optional):

The authors have made good responses and justifications to the reviewer's comments. I am satisfied with the revised manuscript. 
---

## [Editor Report · Acceptance letter]

11 Jul 2023

PONE-D-22-33207R1 

The Burden of Disabilities in Sidama National Regional State, Ethiopia: A Cross-Sectional, Descriptive Study 

Dear Dr. Tenaw:

I'm pleased to inform you that your manuscript has been deemed suitable for publication in PLOS ONE. Congratulations! Your manuscript is now with our production department. 

Kind regards, 

on behalf of

Dr. Chung-Ying Lin 

Academic Editor

PLOS ONE